# Natural Hepacivirus Infection in Tree Shrews: A Call for Routine Screening in Hepatitis Virus Research

**DOI:** 10.3390/v18010027

**Published:** 2025-12-23

**Authors:** Mohammad Enamul Hoque Kayesh, Takahiro Sanada, Michinori Kohara, Kyoko Tsukiyama-Kohara

**Affiliations:** 1Transboundary Animal Diseases Centre, Joint Faculty of Veterinary Medicine, Kagoshima University, Kagoshima 890-0065, Japan; 2Department of Microbiology and Public Health, Faculty of Animal Science and Veterinary Medicine, Patuakhali Science and Technology University, Barishal 8210, Bangladesh; 3Infection Control Unit, Research Center for Infectious Diseases Medical Sciences, Tokyo Metropolitan Institute of Medical Science, Tokyo 156-8506, Japan; sanada-tk@igakuken.or.jp; 4Department of Microbiology and Cell Biology, Tokyo Metropolitan Institute of Medical Science, Tokyo 156-8506, Japan; kohara-mc@igakuken.or.jp

**Keywords:** hepacivirus, tree shrew, natural infection, routine screening, hepatitis virus research

## Abstract

Hepatitis viruses continue to pose major global health challenges, necessitating the development of reliable and well-characterized experimental models. Tree shrews are increasingly recognized as a valuable small animal model because of their natural susceptibility to hepatitis viruses and close phylogenetic relationship with primates. Recent identification of a high prevalence of natural hepacivirus infections in tree shrews underscores the urgent need for routine viral screening of the animals used in hepatitis studies. Undetected infections may confound experimental results, undermine data integrity, and pose risks to laboratory biosecurity. Integrating systematic screening and standardized reporting practices will minimize these risks, enhance reproducibility, and safeguard the integrity of research findings. Moreover, a consistent assessment of the infection status will enhance the translational potential of tree shrews for studying viral hepatitis pathogenesis and evaluating antiviral interventions. This opinion paper emphasizes that ensuring the virological status of tree shrews is not merely a procedural recommendation but also a methodological standard essential for advancing hepatitis virus research.

## 1. Introduction

Animal models are indispensable for advancing our understanding of infectious diseases, particularly for studying viral transmission, pathogenesis of infections, and host immune responses. Progress in the field of hepatitis virus research has long been constrained by the limited availability of suitable models, as chimpanzees have been historically the only species known to be naturally susceptible to both hepatitis B virus (HBV) and hepatitis C virus (HCV). Ethical restrictions and high maintenance costs associated with studies in chimpanzees have further underscored the need for alternative small animal models that accurately recapitulate human infections. Hepaciviruses were once considered human-specific, but the discovery of HCV-like sequences in multiple animal species suggests a wider host range and potential zoonotic transmission [1].

HBV and hepaciviruses are taxonomically and biologically distinct despite both being hepatotropic viruses in humans. HBV is a partially double-stranded DNA virus classified within the family *Hepadnaviridae*, whereas hepaciviruses are positive-sense single-stranded RNA viruses belonging to the family *Flaviviridae*. The genus *Hepacivirus* is a taxonomically defined group of positive-sense single-stranded RNA viruses within the family *Flaviviridae*, as recognized by the International Committee on Taxonomy of Viruses (ICTV). According to the most recent ICTV classification (2024), this genus currently comprises 14 distinct viral species, reflecting the expanding diversity revealed through molecular surveillance and comparative genomics. Viral species are now designated using binomial nomenclature, combining the genus name *Hepacivirus* with a species epithet (e.g., *Hepacivirus hominis*, *Hepacivirus equi*), replacing earlier informal or host-based naming conventions (https://ictv.global/taxonomy; accessed on 18 December 2025). *Hepacivirus hominis*, which includes HCV, is the type species and the most extensively studied member due to its clinical importance in humans. Other species infect a wide range of mammalian hosts, including equids, rodents, bats, non-human primates, and cattle, highlighting substantial genetic diversity and host adaptation within the genus. Adherence to the current ICTV taxonomy is therefore essential to prevent confusion between hepaciviruses and unrelated hepatotropic viruses such as HBV, as well as among formally recognized species, intra-species genotypes, and unclassified hepacivirus-like sequences.

Northern tree shrew (*Tupaia belangeri*), also known as tupaia, has emerged as a promising alternative model because of its natural susceptibility to HBV and HCV [2]. Genomic analyses of the northern tree shrew revealed its close evolutionary relationship with primates, indicating a possibility of its use as a mammalian model for studying human diseases [3,4]. In addition to its genetic closeness to primates, the tree shrew possesses physiological and immunological characteristics, including conserved signaling pathways involved in innate and adaptive immune responses, as well as comparable patterns of hepatic gene expression during viral infection, which mirror key aspects of human biology and further enhance its value in biomedical research, making it uniquely valuable for investigating viral infections and assessing potential therapeutics [5,6,7].

Sodium taurocholate co-transporting polypeptide, a functional HBV entry receptor, was identified by its interaction with the viral pre-S1 domain in primary tupaia hepatocytes [8]. Recently, transplantation of primary tupaia hepatocytes into chimeric mice has enabled efficient HBV infection, providing a valuable in vivo model [9]. Host genetic variation plays a critical role in determining the outcome of hepacivirus infection. For example, Norway rat hepacivirus infection outcomes vary across mouse strains, ranging from viral clearance to chronicity [10]. Recent studies investigating species barriers to HCV infection have identified murine liver factors such as CD302 (also known as Dcl-1 and Clec13a) and CR1L (complement component 3b/4b receptor 1–like, also known as Crry) as key restrictors of viral propagation, highlighting interspecies differences in liver-intrinsic antiviral immunity [11]. In contrast, tree shrews express several human-like factors essential for HCV entry and replication, including cluster of differentiation 81 (CD81), occludin (OCLN), microRNA-122 (miR-122), SEC14-like lipid binding 2 (SEC14L2), and Niemann–Pick C1-like 1 (NPC1L1), all of which are highly homologous to their human counterparts [12,13,14], reinforcing the value of this species as a small animal model for hepatitis research.

Tree shrews have gained attention in studies of viral hepatitis infections, offering insights into the pathogenesis of both HBV and HCV. Their utility is particularly evident in addressing the challenges posed by the ‘stealth’ nature of the HBV, which allows the virus to evade early innate immune detection [15]. In acute HBV infection, tree shrews show hallmark features observed in humans, including viral replication, covalently closed circular DNA formation, seroconversion, elevation of alanine aminotransferase levels, and liver pathology. Early antiviral responses in tree shrews at 14 days post-infection include activation of Toll-like receptors TLR1–TLR9 and cyclic-GMP-AMP synthase, expression of interferon β mRNA, and downregulation of PDL1 [16,17], supporting their suitability as an acute infection model.

This model also reflects the progression of chronic HBV infection, including the development of hepatocellular carcinoma, which parallels the course of human disease [18]. Human HBV is efficiently transmitted through multiple generations in tree shrews, reflecting their high sensitivity to HBV infection [19]. Chronic infection in tree shrews has been associated with impaired interferon signaling—in particular, reduced interferon β expression—similar to that observed in patients with chronic HBV infection [20,21]. Histopathological features such as steatosis, hepatitis, cirrhotic nodules, and tumorigenesis have been consistently reported in studies of HBV and HCV using this model [22,23,24,25,26]. Notably, neonatal HBV infection in tree shrews often progresses to chronic infection accompanied by histopathological alterations in the liver that closely resemble those observed in human HBV infections [18,21,26]. Similarly, in HCV infection, histopathological changes in the liver in tree shrews are comparable to those observed in humans [22,23]. Moreover, HCV has been shown to undergo genetic evolution, particularly within the hypervariable region 1 (HVR1) of the envelope protein, during infection in primary tupaia hepatocytes, similar to its changes in humans and chimpanzees [25]. In addition to pathogenesis research, tree shrews have supported therapeutic advancements, including the CRISPR/Cas9-based targeting of HBV covalently closed circular DNA [27,28] and vaccine development [29]. Hepatitis E virus (HEV) is the leading cause of acute hepatitis globally, with a particularly high prevalence in developing regions [30,31]. In the 2024 ICTV taxonomy update, HEV is formally designated *Paslahepevirus balayani* in the family *Hepeviridae*, encompassing eight genotypes (HEV-1 to HEV-8), of which genotypes 1–4 are most commonly associated with human infection (https://ictv.global/taxonomy; accessed on 18 December 2025). Recent studies have demonstrated that tree shrew, already established as a model for HBV and HCV research, is also susceptible to HEV infection, supporting its utility in studies of this type of hepatitis [32]. HEV infection during pregnancy is associated with severe outcomes, including acute liver failure, miscarriage, and preterm delivery. However, the mechanisms underlying this heightened pathogenicity remain poorly understood. Studies employing pregnant tree shrews have shown that HEV, particularly genotype 4, replicates in multiple organs, including the uterus, liver, intestine, brain, spinal cord, and bile, with the highest viral titers detected in bile, followed by the uterus and liver, and the lowest viral loads observed in the spinal cord. Elevated estrogen levels may promote viral replication and exacerbate disease severity [33]. Consistent with this, HEV genotype 4 infection in tree shrew liver induced typical viral hepatitis lesions, including disorganized hepatic cords, perivascular lymphocytic infiltration, hepatocyte loss around the central vein, Kupffer cell proliferation, sinusoidal dilation, and hepatocyte atrophy, reflecting significant hepatic damage. Notably, abortions were observed in two HEV-infected tree shrews, and stillbirths occurred at a rate of 33% in HEV-infected group of pregnant tree shrews [33]. Given the recent discovery of natural hepacivirus infection in tree shrews, routine screening is imperative to avoid confounding experimental outcomes and preserve the integrity of hepatitis virus research.

## 2. Detection of Hepaciviruses in Tree Shrews and Other Animals

HCV infection remains a major global health concern, affecting approximately 3% of the human population. However, its zoonotic origins remain unresolved [34]. Hepaciviruses, the closest known relatives of HCV, have been identified in multiple animal species, including bats, cattle, horses, dogs, rats, and ducks. Although the degree of chronicity varies across hosts, hepaciviruses consistently show liver tropism and share key pathological features with HCV [35,36,37,38,39,40]. Equine hepacivirus (EqHV) (*Hepacivirus equi*), formerly known as non-primate hepacivirus, is a hepatotropic member of the *Flaviviridae* family that infects horses and possibly dogs and is the closest known relative of human HCV (https://ictv.global/taxonomy; accessed on 18 December 2025). EqHV is detected in horses and can cause both acute and chronic liver disease, exhibiting genetic and clinical similarities to HCV; it frequently results in subclinical infection and only occasionally progresses to chronicity [41,42,43,44,45,46]. Its translation is stimulated by miR-122 and the 3′-UTR, and the viral NS3-4A protease inactivates innate immune signaling through cleavage of mitochondrial antiviral-signaling protein, mirroring HCV immune evasion strategy [39,44]. These characteristics highlight the close molecular and pathogenic relationship between EqHV and HCV, and underscore the value of studying hepaciviruses in diverse nonprimate hosts [39]. EqHV has been reported to cross-infect donkeys under experimental conditions, suggesting that donkeys may serve as natural hosts for EqHV [47]. Recent discoveries of diverse and divergent hepaciviruses in rodents have expanded our understanding of their evolutionary history and highlighted the potential role of small mammals as natural reservoirs [48]. In Collaborative Cross mice, rodent hepacivirus infection produces strain-dependent liver pathology, ranging from transient hepatitis to persistent infection with sustained inflammation and occasional fibrosis. These host-genetically driven hepatic outcomes mirror key features of human HCV disease, reinforcing the relevance of rodent hepaciviruses when considering the evolutionary diversity and pathogenic potential of hepaciviruses circulating in small-mammal reservoirs [10].

Canine hepacivirus (CHV), the closest known genetic relative of human HCV, has been identified in domestic dogs and horses [40,49]. Although CHV is a promising model for HCV research, its hepatotropism and involvement in canine liver diseases remain unclear [50]. Multiple studies have found no association between CHV infection and chronic liver disease in dogs, as viral RNA was undetectable in liver samples from affected and control animals across UK and European cohorts [50,51]. Interestingly, the CHV NS3/4A protease has been shown to cleave human MAVS and TRIF, key components of the innate antiviral signaling pathway, similar to its role in HCV, suggesting a conserved mechanism of immune evasion and possible evolutionary link between the two viruses [49].

Tree shrew hepacivirus is a naturally occurring hepatotropic virus belonging to diverse genus *Hepacivirus*, the members of which infect a wide range of mammalian hosts [36]. Using published RNA sequencing data, Misfud et al. identified a novel hepacivirus sequence in northern tree shrew [52]. Recent metagenomic analyses from the Yunnan Province of China have also revealed extensive viral diversity in wild tree shrews, identifying genomic fragments from 18 viral families, including novel hepaciviruses and hepeviruses. These findings place tree shrew–associated hepaciviruses within the broader small-mammal virome, contextualizing their relevance for viral ecology and host–pathogen co-evolution, and suggest that northern tree shrews may serve as a natural reservoir for hepatitis-related viruses [53]. Another study has detected hepaciviruses in wild tree shrews in China [54]. Notably, a recent report from Japan described high prevalence of tree shrew hepaciviruses 1 (70.3%) and 2 (40.5%) in laboratory-maintained tree shrews, originally obtained from the Kunming Institute of Zoology, China [36]. Although most tree shrews came from this single geographic location (Kunming, Yunnan Province, China), differences in housing conditions, husbandry practices, and population sizes may exist, limiting the generalizability of these findings to other colonies. Owing to their close phylogenetic relationship with humans and wide geographical distribution, tree shrews are considered potential reservoirs of various zoonotic pathogens [54,55]. The detection of natural hepacivirus infections in tree shrews raises important questions regarding host adaptation, viral persistence, and potential for interspecies transmission. Considered alongside reports of naturally occurring hepaciviruses in other nonhuman species, these findings highlight the complex ecology of hepaciviruses and their evolutionary links with human HCV. Collectively, this evidence underscores a critical need for routine viral screening in animal models, particularly tree shrews, used in hepatitis research, to ensure data reliability and biosafety.

## 3. Necessity of Routine Hepacivirus Screening in Tree Shrews for Hepatitis Research

The high prevalence of hepaciviruses in tree shrews is particularly significant because of their increasing use as animal models for hepatitis research. However, liver histopathology and clinical parameters in naturally infected tree shrews remain only partially characterized and are largely inferred from experimental infections, highlighting an important gap for future systematic investigation. Together, these findings raise concerns regarding undetected pre-existing infections in both wild-caught and laboratory-maintained populations. Without proper virological screening, hepacivirus infections may interfere with viral replication dynamics, confound experimental outcomes, and compromise data integrity, particularly in studies involving human hepatitis viruses such as HBV, HCV, and HEV; nonetheless, the tree shrew model may also be applicable to broader liver-related viral studies where appropriate [53]. Latent or subclinical infections may remain unnoticed, potentially affecting liver pathology data, immune responses, and efficacy of tested antiviral treatments. Moreover, undiagnosed infections can contribute to the pronounced individual variability often observed across studies, thereby complicating interpretation and reproducibility. Therefore, routine pre-screening is essential to ensure the consistency, validity, and reliability of experimental designs. For prescreening, serum and fecal samples are preferred, as they allow detection of pathogen-specific antigens, antibodies, or nucleic acids (e.g., RNA), as appropriate [32]. Although liver biopsy is possible [56], it is not preferred. Sample collection is ideally performed upon arrival and prior to the initiation of experiments involving tree shrews and, where applicable, at multiple time points throughout the study. Standardized and validated protocols for hepacivirus detection, including RT-PCR and serological assays for prescreening, should be consistently applied in studies involving tree shrews, particularly those focused on hepatitis research. Beyond research implications, natural infections may also affect the general health and welfare of animals, which can further influence experimental readouts. Finally, potential biosafety concerns for laboratory personnel must not be overlooked because unrecognized infections may pose hidden hazards. Thus, implementing comprehensive and systematic screening is critical not only for maintaining scientific rigor and reproducibility but also for upholding ethical and biosafety standards in hepatitis virus research.

## 4. Conclusions

Identification of natural hepacivirus infections in tree shrews underscores the critical need for routine viral screening in hepatitis research. Integrating systematic screening into experimental protocols will minimize confounding variables, improve data reliability, and protect research integrity and laboratory biosecurity. To fully harness the translational and comparative value of tree shrews, the research community must adopt standardized detection, monitoring, and reporting practices. Given their high susceptibility to hepatitis viruses and close phylogenetic relationships with primates, tree shrews represent a uniquely relevant and biologically informative model for studying the pathogenesis of viral hepatitis and evaluating antiviral therapies. Ensuring the virological status of these animals is not only a methodological necessity but also a foundational prerequisite for scientific rigor and advancement of hepatitis virus research.

## Data Availability

The contributions presented in this study are included in the article.

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
