# Peer review of "Natural Hepacivirus Infection in Tree Shrews: A Call for Routine Screening in Hepatitis Virus Research"

_viruses, 2025, doi:10.3390/v18010027_

Round 1
Reviewer 1 Report
Comments and Suggestions for Authors
Tree shrews can replace non-human primates as an important experimental animal in many ways. Tree shrews of experimental animals can be purchased from Kunming Institute of Zoology, Chinese Academy of Sciences. Therefore, this article proposes that it is very important to conduct examinations for relevant pathogens, especially hepatitis C virus and hepatitis E virus, before conducting relevant experimental animal models.
Q1. What is the difference between hepatitis B virus, hepaciviruses, and Hepacivirus? Currently, Hepacivirus is a genus. According to the latest 2024 classification by the ICTV, this genus comprises 14 distinct viruses (https://ictv.global/taxonomy). Viral species are now named using binomial nomenclature, combining the genus name with the species name. Examples include Hepacivirus hominis, Hepacivirus bovis, etc. It is recommended that this article provide an appropriate explanation of the current classification within this genus.
Q2. Line 35-36: “ hepatitis B (HBV) and hepatitis C (HCV) viruses.” should be changed to hepatitis B virus (HBV) and hepatitis C virus (HCV). Hereafter, hepatitis B virus and hepatitis C virus will be abbreviated as HBV and HCV. Line 95: hepatitis B and C virus research, revised “HBV and HCV”.
Q3. Line 132: The gene Hepacivirus should be changed to italicized Hepacivirus
Q4. Line 137-138: “hepatitis E virus”should be changed to “hepeviruses”.
Author Response
Tree shrews can replace non-human primates as an important experimental animal in many ways. Tree shrews of experimental animals can be purchased from Kunming Institute of Zoology, Chinese Academy of Sciences. Therefore, this article proposes that it is very important to conduct examinations for relevant pathogens, especially hepatitis C virus and hepatitis E virus, before conducting relevant experimental animal models.
Q1. What is the difference between hepatitis B virus, hepaciviruses, and Hepacivirus? Currently, Hepacivirus is a genus. According to the latest 2024 classification by the ICTV, this genus comprises 14 distinct viruses (https://ictv.global/taxonomy). Viral species are now named using binomial nomenclature, combining the genus name with the species name. Examples include Hepacivirus hominis, Hepacivirus bovis, etc. It is recommended that this article provide an appropriate explanation of the current classification within this genus.
Response: We thank the reviewer for this insightful comment and for highlighting the importance of using the current ICTV classification and nomenclature.
Hepatitis B virus (HBV) and hepaciviruses belong to entirely different viral families and are taxonomically and biologically distinct. HBV is a partially double-stranded DNA virus classified within the family Hepadnaviridae, whereas hepaciviruses are positive-sense single-stranded RNA viruses classified within the family Flaviviridae. Thus, HBV is not related to hepaciviruses despite both being hepatotropic viruses in humans. Hepaciviruses refer to viruses within the genus Hepacivirus. As correctly noted by the reviewer, Hepacivirus is a genus, not a single virus. According to the 2024 ICTV taxonomy, the genus Hepacivirus comprises 14 recognized viral species, which are now named using binomial nomenclature that combines the genus name (Hepacivirus) with a species epithet (e.g., Hepacivirus hominis, Hepacivirus bovis). The virus commonly referred to as hepatitis C virus corresponds to the species Hepacivirus hominis.
In response to the reviewer’s recommendation, we have revised the manuscript to explicitly distinguish between HBV and hepaciviruses, clarify that Hepacivirus is a genus, and describe the current ICTV (2024) classification and binomial species nomenclature within this genus. A brief explanatory paragraph has been added to ensure taxonomic accuracy and to avoid potential confusion for readers (lines 41–60).
Q2. Line 35-36: “ hepatitis B (HBV) and hepatitis C (HCV) viruses.” should be changed to hepatitis B virus (HBV) and hepatitis C virus (HCV). Hereafter, hepatitis B virus and hepatitis C virus will be abbreviated as HBV and HCV. Line 95: hepatitis B and C virus research, revised “HBV and HCV”.
Response: Thank you for the helpful suggestion. We have revised the terminology throughout the manuscript to use ‘hepatitis B virus (HBV)’ and ‘hepatitis C virus (HCV)’ at first mention (lines 35–36), and thereafter abbreviated them as HBV and HCV. In addition, the phrase ‘hepatitis B and C virus research’ has been corrected to ‘HBV and HCV research’ (line 122).
Q3. Line 132: The gene Hepacivirus should be changed to italicized Hepacivirus
Response: Thank you for pointing this out. We have corrected by italicizing Hepacivirus in accordance with standard taxonomic formatting (line 186).
Q4. Line 137-138: “hepatitis E virus”should be changed to “hepeviruses”.
Response: Thank you for the suggestion. We have revised it by replacing “hepatitis E virus” with “hepeviruses,” as recommended (line 191).
Reviewer 2 Report
Comments and Suggestions for Authors
This manuscript discusses an important and often overlooked methodological confounding factor in hepatitis research: the presence of natural hepacivirus infections in tree shrew. Given the increasing use of tree shrews as a bridge model between rodents and non-human primates, the authors’ call for standardized virological screening is both timely and essential. The manuscript is well-written and provides actionable recommendations. I have the following concerns need to be addressed.
1. While the authors correctly advocate for "standardized protocols," it would be better if the authors can briefly delineate a recommended "Minimum Screening Panel." This should specify preferred sample matrices, the necessity of NGS-based discovery vs. targeted RT-qPCR et al.
2. The term “PTHs” (primary tree shrew hepatocytes) is introduced early but underutilized. I suggest either consistent usage or reverting to the full term to maintain flow.
3. In Section 2, a brief sentence linking these findings to the broader "virome" of small mammals would help contextualize the tree shrew findings within the larger field of viral ecology and host-pathogen co-evolution.
Author Response
This manuscript discusses an important and often overlooked methodological confounding factor in hepatitis research: the presence of natural hepacivirus infections in tree shrew. Given the increasing use of tree shrews as a bridge model between rodents and non-human primates, the authors’ call for standardized virological screening is both timely and essential. The manuscript is well-written and provides actionable recommendations. I have the following concerns need to be addressed.
- While the authors correctly advocate for "standardized protocols," it would be better if the authors can briefly delineate a recommended "Minimum Screening Panel." This should specify preferred sample matrices, the necessity of NGS-based discovery vs. targeted RT-qPCR et al.
Response: We thank the reviewer for this insightful comment. In response, we have revised the text to clarify a practical prescreening approach centered on targeted RT-PCR and serology-based assays and have removed references to NGS, as it is not routinely feasible for prescreening purposes (lines 243–246).
- The term “PTHs” (primary tree shrew hepatocytes) is introduced early but underutilized. I suggest either consistent usage or reverting to the full term to maintain flow.
Response: We thank the reviewer for this helpful comment. In response, we have revised the manuscript to remove the abbreviation “PTHs” and consistently use the full term primary tupaia hepatocytes to improve clarity and maintain narrative flow (line 73).
- In Section 2, a brief sentence linking these findings to the broader "virome" of small mammals would help contextualize the tree shrew findings within the larger field of viral ecology and host-pathogen co-evolution.
Response: We thank the reviewer for this insightful comment. In response, we have added a brief sentence in Section 2 (lines 191–194) linking tree shrew–associated hepaciviruses to the broader small-mammal virome, thereby contextualizing these findings within viral ecology and host–pathogen co-evolution.
Reviewer 3 Report
Comments and Suggestions for Authors
In this manuscript, Kayesh and colleagues summarize the growing body of evidence on hepaciviruses in tree shrews and other animal hosts, emphasizing the implications of natural infections for the use of tree shrews as models in hepatitis virus research. They highlight the high prevalence of tree shrew hepaciviruses in laboratory colonies, review the broader ecology of hepaciviruses in diverse mammals, and argue that undetected infections may confound pathogenesis and vaccine/antiviral studies and raise biosafety concerns. The article is clearly written and timely, and the overall message is relevant to the field.
However, in my opinion the manuscript would benefit from some revisions before it can be accepted. I would recommend minor revision, mainly aimed at making the routine screening more operational and reducing some redundancy.
The central message of the paper is important, but it is formulated in general terms. I strongly suggest that the authors translate the conceptual recommendation into more practical guidance. For example, they could propose:
Preferred sample types (e.g., serum, liver biopsies, faeces) and timing (on arrival, pre-experiment, longitudinally).
A minimal panel of tests (e.g., RT-PCR on serum and/or liver, serology, and when NGS is realistically indicated) and their intended purpose (screening vs. confirmatory).
The sections describing detection of hepaciviruses in tree shrews and other mammals are informative but largely descriptive. It would be useful to more explicitly discuss:
Sample sizes, geographic origin and housing conditions of the tree shrew cohorts in which high prevalences were reported, and the limitations of extrapolating to all laboratory colonies.
Whether liver histopathology and clinical parameters in naturally infected tree shrews have been systematically characterized or remain largely inferred from experimental infections and other hepacivirus models.
At several points, “hepatitis virus research” is used broadly; please clarify whether the recommendation is primarily intended for HBV/HCV/HEV studies, or more generally for any liver-related viral research using tree shrews.
When referring to “hepaciviruses in other animals”, it may help to briefly indicate whether those infections have been associated with overt liver disease or mainly subclinical infection.
Precision in examples
When citing the work on pregnant tree shrews infected with zoonotic HEV, please specify the HEV genotype(s) used and the key pathological findings (e.g., organ distribution, pregnancy outcomes).
Author Response
In this manuscript, Kayesh and colleagues summarize the growing body of evidence on hepaciviruses in tree shrews and other animal hosts, emphasizing the implications of natural infections for the use of tree shrews as models in hepatitis virus research. They highlight the high prevalence of tree shrew hepaciviruses in laboratory colonies, review the broader ecology of hepaciviruses in diverse mammals, and argue that undetected infections may confound pathogenesis and vaccine/antiviral studies and raise biosafety concerns. The article is clearly written and timely, and the overall message is relevant to the field.
However, in my opinion the manuscript would benefit from some revisions before it can be accepted. I would recommend minor revision, mainly aimed at making the routine screening more operational and reducing some redundancy.
The central message of the paper is important, but it is formulated in general terms. I strongly suggest that the authors translate the conceptual recommendation into more practical guidance. For example, they could propose:
Preferred sample types (e.g., serum, liver biopsies, faeces) and timing (on arrival, pre-experiment, longitudinally).
Response: We thank the reviewer for this comment. We have included this information in the revised manuscript (lines 238–243).
A minimal panel of tests (e.g., RT-PCR on serum and/or liver, serology, and when NGS is realistically indicated) and their intended purpose (screening vs. confirmatory).
Response: We thank the reviewer for this insightful comment. In response, we have revised the text to clarify a practical prescreening approach centered on targeted RT-PCR and serology-based assays and have removed references to NGS, as it is not routinely feasible for prescreening purposes (lines 243–246).
The sections describing detection of hepaciviruses in tree shrews and other mammals are informative but largely descriptive. It would be useful to more explicitly discuss:
Sample sizes, geographic origin and housing conditions of the tree shrew cohorts in which high prevalences were reported, and the limitations of extrapolating to all laboratory colonies.
Response: We thank the reviewer for this comment. We have revised the manuscript based on the reviewer comments (lines 197–200).
Whether liver histopathology and clinical parameters in naturally infected tree shrews have been systematically characterized or remain largely inferred from experimental infections and other hepacivirus models.
Response: We thank the reviewer for raising this point. We have clarified in the revised manuscript that liver histopathology and clinical parameters in naturally infected tree shrews remain only partially characterized and are largely inferred from experimental infections, highlighting an important gap for future systematic investigation (line 212–215).
At several points, “hepatitis virus research” is used broadly; please clarify whether the recommendation is primarily intended for HBV/HCV/HEV studies, or more generally for any liver-related viral research using tree shrews.
Response: We thank the reviewer for this comment. We have revised the text to clarify that the recommendation primarily pertains to hepatitis virus research focused on hepatotropic viruses (including HBV, HCV, and HEV), while noting that the tree shrew model may also be applicable to broader viral studies where appropriate (Sanada, et al., Virology, 529, 101–110; Kayesh et al., Viruses, 13(8), 1641) (lines 219–221).
When referring to “hepaciviruses in other animals”, it may help to briefly indicate whether those infections have been associated with overt liver disease or mainly subclinical infection.
Response: We thank the reviewer for this suggestion. We have revised the manuscript to clarify the clinical outcomes of hepacivirus infections in other animals (lines 149–151; 169–175).
Precision in examples
When citing the work on pregnant tree shrews infected with zoonotic HEV, please specify the HEV genotype(s) used and the key pathological findings (e.g., organ distribution, pregnancy outcomes).
Response: We thank the reviewer for the suggestion. We have now specified that the studies involved HEV genotype 4 infection in tree shrews and have included key pathological findings, such as viral replication across multiple organs, liver histopathological abnormalities, and effects on pregnancy outcomes (lines 127–136).